# Design of a mmWave Antenna Printed on a Thick Vehicle-Glass Substrate Using a Linearly Arrayed Patch Director and a Grid-Slotted Patch Reflector for High-Gain Characteristics

**DOI:** 10.3390/s22166187

**Published:** 2022-08-18

**Authors:** Changhyeon Im, Tae Heung Lim, Hosung Choo

**Affiliations:** 1Department of Electronic and Electrical Engineering, Hongik University, Seoul 04066, Korea; 2Department of Electrical Engineering, Ulsan National Institute of Science and Technology (UNIST), Ulsan 44919, Korea

**Keywords:** linearly arrayed patch director, grid-slotted patch reflector, glass antenna, high-gain, 5G vehicle application

## Abstract

This paper proposes a 5G glass antenna that can be printed on the thick window glass of a vehicle. The proposed antenna consists of a coplanar waveguide (CPW), a printed monopole radiator, parasitic elements, a linearly arrayed patch director, and a grid-slotted patch reflector. The linearly arrayed patch director and grid-slotted patch reflector are applied to improve the bore-sight gain of the antenna. To verify the performance improvement and feasibility, the proposed antenna is fabricated, and a reflection coefficient and a radiation pattern are measured and compared with the simulation results. The measured reflection coefficient shows broadband characteristics of less than −10 dB from 24.1 GHz to 31.0 GHz (fractional bandwidth of 24.6%), which agrees well with the simulation results. The reflection coefficients are −33.1 dB by measurement and −25.7 dB by simulation, and the maximum gains are 6.2 dBi and 5.5 dBi at 28 GHz, respectively. These results demonstrate that the proposed antenna has high-gain characteristics being suitable for 5G wireless communications.

## 1. Introduction

With the recent development of autonomous vehicle technologies, considerable efforts are devoted to employing 5G wireless communication systems for vehicle applications [1,2,3,4,5,6]. 5G vehicle wireless communication systems essentially require mmWave antennas that can stably maintain vehicle wireless communication links according to the signal transmission/reception environments while the vehicle is driving. To meet such requirements, many studies on the mmWave vehicle antennas have investigated by employing multiple antennas inside or outside vehicles [7,8,9,10,11,12], using a roof-top antenna for the coupling reduction with the vehicle body [13,14,15], and mounting the antennas inside the shark-fin radome [16,17]. These antennas have good impedance matching and high-gain characteristics for improving the reliability of 5G signal reception. However, it is difficult to practically use such antenna design techniques in vehicle applications because the change in the vehicle structures creates enormous costs in the vehicle manufacturing process. To resolve these problems, research on low-cost mmWave antennas that print various shapes such as a rectangular patch, a helical antenna, and a slot antenna on glass substrates, has been carried out [18,19,20]. Nevertheless, these antennas generally adopt thin glass substrates to reduce pattern distortion and radiation loss, and it is still difficult to apply them to thick glass substrates of actual vehicles. Although some mmWave antennas printed on very thick vehicle-glass substrates have been recently introduced, these vehicle-glass antennas generally exhibit low bore-sight gain characteristics [21,22,23].

In this paper, we propose an mmWave antenna printed on thick vehicle glass with a linearly arrayed patch director and a grid-slotted patch reflector for high-gain characteristics. The proposed antenna consists of a coplanar waveguide (CPW), a printed monopole radiator, parasitic elements, the linearly arrayed patch director, and the grid-slotted patch reflector. The monopole radiator with the parasitic elements is printed on the top surface of the thick vehicle glass. In addition, the linearly arrayed patch director is designed on the same surface with the monopole using seven rectangular patch elements to enhance the bore-sight gain. On the other side, the grid-slotted patch reflector, which consists of 5 × 4 rectangular patches, is printed to further improve the bore-sight gain and reduce the pattern distortion. This reflector can operate as a frequency selective surface (FSS), which reflects electromagnetic waves at a specific frequency [24,25]. The critical design parameters of the proposed antenna are then optimized using the CST Studio Suite EM simulation software [26] and re-validated by HFSS [27] and FEKO EM simulation software tool [28]. To verify the feasibility, it is fabricated and measured in a full anechoic chamber to obtain the antenna characteristics, such as reflection coefficients, gains, and radiation patterns. These results demonstrate that the proposed antenna on the thick vehicle glass has high-gain characteristics being suitable for 5G wireless communications.

## 2. Proposed Antenna Design

Figure 1 shows the geometry of the proposed on-glass antenna for the vehicle applications. The proposed antenna consists of a CPW transmission line, a printed monopole radiator, parasitic elements, a linearly arrayed patch director, and a grid-slotted patch reflector. Herein, the geometry and concept of the proposed antenna are modeled based on the previous research [23]. The CPW transmission line has an inner conductor with a length and a width of *l*_1_ and *w*_4_, and the rectangular grounds (*l*_1_ × *w*_5_) are designed on both sides of the inner line having a gap of *g*_3_. This CPW transmission line is printed on the top of a thick vehicle-glass substrate (*ε_r_* = 6.9, *tanδ* = 0.03 at 28 GHz) having a thickness of *t*. Note that if the vehicle-glass substrate thickness becomes thicker, it is difficult to use the conventional monopole antenna due to performance degradations such as pattern distortion and gain reduction, which will be specifically discussed in Section 3. The CPW transmission line is usually less affected by the substrate thickness compared to other transmission lines (i.e., a microstrip line, a strip line, and a CPW with a ground), so that it is advantageous for use as a feeder in the proposed antenna employing the thick vehicle-glass substrate. The printed monopole radiator has a length of *l*_3_ and a width of *w*_4_ and is directly connected to the CPW inner conductor [29,30,31]. Four identical parasitic elements with a rectangular patch shape (*l*_2_ × *w*_3_) are located at distances of *g*_2_ from the CPW ground and *g*_3_ from the monopole radiator. The parasitic elements are indirectly coupled by electromagnetic fields from the monopole radiator, and surface current distributions near the parasitic elements can enlarge an effective aperture resulting in the improved radiation efficiency. On the same surface as the monopole and parasitic elements, seven rectangular patches having a width of *w*_2_ are linearly arrayed with an identical interval of *g*_1_ to model a director that can reduce the pattern distortion by focusing the electromagnetic fields at the bore-sight direction. The grid-slotted patch reflector with an *M* × *N* rectangular configuration is printed on the bottom of the thick vehicle-glass substrate, where *M* and *N* represent the number of reflector patches along the *x*- and *y*-axis, respectively. Each patch has a width *w*_6_ and an equal distance *g*_4_ between the adjacent patches, and the reflector can more improve the bore-sight gain by reflecting the back radiation field to make a constructive interference at the main lobe direction. The proposed antenna is modeled, and its critical design parameters are optimized using the CST Studio Suite EM simulator. The optimized parameters are listed in Table 1.

## 3. Analysis

Figure 2 presents the reflection coefficients according to the distance *d* from the end edge of the CPW ground to the starting edge of the grid-slotted patch reflector along the *x*-axis. The resonance of the proposed antenna can be adjusted by the overlapping area between the reflector and the printed monopole radiator with the parasitic elements, since the overlapping area can confine the strong electromagnetic fields inside the thick vehicle-glass substrate. Figure 3 illustrates the maximum gain and the main lobe direction *θ*_0_ in the upper hemisphere of the radiation pattern in accordance with a variation of *d*. The resulting maximum gain shows a peak level at the optimized distance of *d* = 1.06 mm, which allows the main lobe direction of *θ*_0_ to be approximately 0°.

Figure 4 shows the reflection coefficients according to the printed monopole length *l*_3_. The resonance frequency increases when *l*_3_ varies from 1.63 mm to 2.43 mm since the overlapping area between the printed monopole radiator and the reflector become larger.

To observe antenna characteristics according to the shape of the director, the bore-sight gain is simulated by changing the number of patch elements along the *x*- and *y*-axis, as shown in Figure 5. *L_x_* and *L_y_* specify the number of the patches in the *x*- and *y*-axis, respectively. High bore-sight gains are obtained when *L_x_* is 1, which means the director is a linear array, and *L_y_* of 7 is decided to have the maximum bore-sight gain of 5.5 dBi. Figure 6 shows the simulated bore-sight gains according to *g*_1_ and *w*_2_, while *L_x_* and *L_y_* are fixed to be 1 and 7. The bore-sight gains are examined in the parameter ranges of 0.1 mm ≤ *g*_1_ ≤ 1 mm and 1.5 mm ≤ *w*_2_ ≤ 2.5 mm. The peak gain of 5.5 dBi is obtained when the optimal values of *g*_1_ and *w*_2_ are 0.5 mm (0.125λ_0_) and 1.9 mm (0.48λ_0_), where λ_0_ is the wavelength at the operating frequency of 28 GHz.

Figure 7 illustrates the bore-sight gain in accordance with the gap (*g*_4_) and the width (*w*_6_) of the grid-slotted patch reflector. For a fixed *M* = 5 and *N* = 4, the bore-sight gain is observed in the ranges of 0.1 mm ≤ *g*_4_ ≤ 1 mm and 1.5 mm ≤ *w*_6_ ≤ 2.5 mm. Figure 8 shows the reflection coefficient and transmission coefficient of the grid-slotted patch reflector. The reflection coefficient is −1.7 dB, and the transmission coefficient is −10.1 dB at 28 GHz. These results mean that electromagnetic waves of 28 GHz are reflected well, thereby increasing the bore-sight gain [24,32,33].

Figure 9a,b present the maximum gain and the main lobe direction *θ*_0_ of the conventional printed monopole antenna in comparison with the proposed antenna according to the glass thickness *t*. For the conventional monopole antenna, the trends of the maximum gain and main lobe direction are considerably irregular. In particular, at a conventional vehicle glass thickness of 3.2 mm, *θ*_0_ is significantly tilted more than 150° from the bore-sight direction. On the other hand, for the proposed antenna, the main lobe direction *θ*_0_ is less than 10° in the range of 0.8 mm ≤ *t* ≤ 1.1 mm and 2.75 mm ≤ *t* ≤ 3.5 mm. In Figure 10, gain characteristics of the proposed antenna are observed according to variations of *w*_1_ from 2.5λ_0_ to 15λ_0_. The bore-sight gain fluctuates slightly when *w*_1_ is less than 5λ_0_, and it saturates to 4.5 dBi when *w*_1_ is more than 10λ_0_. The results confirm that the proposed antenna can be suitable for use in the large vehicle glass.

## 4. Fabrication and Measurement

Figure 11a,b shows the fabricated mmWave on-glass antenna with the linearly arrayed patch director and the grid-slotted patch reflector. The proposed antenna is fabricated using a thermosetting adhesive to strongly attach the printed antenna copper layers to the glass substrate. The proposed antenna is fed by a K-type (2.92 mm) connector operating at up to 40 GHz. The inner and outer conductors of the connector are electrically connected to the inner conductor and rectangular grounds of the CPW line by soldering. Figure 11c shows a photograph of the measurement setup, and antenna characteristics such as reflection coefficients and radiation patterns are measured in the full anechoic chamber.

As shown in Figure 12, the measured reflection coefficient shows broadband characteristics of less than −10 dB from 24.1 GHz to 31.0 GHz (fractional bandwidth of 24.6%), which agrees well with the simulation results. At 28 GHz, the measured reflection coefficient is −33.1 dB, while simulated reflection coefficients are −25.7 dB by CST, −23.8 dB by HFSS, and −27.9 dB by FEKO. Figure 13 illustrates maximum gains with or without the linearly arrayed patch director and the grid-slotted patch reflector. At 28 GHz, simulated result of the proposed antenna shows an improved maximum gain of 5.52 dBi. It is higher than the gain of 1.9 dBi without the reflector and the gain of 2.9 dBi without the director. The measurement results also agree well with the simulation, and the maximum measured gain is 6.20 dBi at 28 GHz. The radiation efficiency of the proposed antenna is 39%, which is higher than 31% without the reflector and 38% without the director.

Figure 14 shows the current distributions of the proposed antenna without the director and without the reflector. When the grid-slotted patch reflector is employed, the current distributions of the reflector are increased at 28 GHz.

Figure 15 shows two-dimensional (2-D) radiation patterns in *zx*- and *zy*-planes at 28 GHz. The measured maximum gain in the *zx*-plane is 6.20 dBi at *θ* = 14°, and the simulated maximum gain is 5.54 dBi at *θ* = −2° by CST. In the *zy*-plane, the measured maximum gain is 3.24 dBi at *θ* = −3°, and simulated result is 5.52 dBi at *θ* = 0° by CST. Figure 16 illustrates simulated 3-D radiation patterns according to the presence of the director and the reflector. The resulting 3-D radiation beam shapes obviously show that the bore-sight gain is improved by adding both the reflector and the director.

## 5. Conclusions

We proposed an mmWave on-glass antenna with a linearly arrayed patch director and a grid-slotted patch reflector. The proposed antenna was printed on the thick vehicle-glass substrate, which did not require the remodeling of the vehicle structures. To improve the bore-sight gain, the linearly arrayed patch director and the grid-slotted patch reflector were employed. The director and reflector were optimized to achieve high-gain characteristics on the thick glass substrate. The measured reflection coefficient of the proposed antenna was −33.1 dB, and the maximum gain was 6.2 dBi at 28 GHz. In addition, it was confirmed that the bore-sight gain was saturated to 4.5 dBi even when the size of the glass window was increased. Through the results, we verified that the proposed antenna achieving the high-gain characteristics could be suitable for 5G wireless communications.

## Figures and Tables

**Figure 1 sensors-22-06187-f001:**
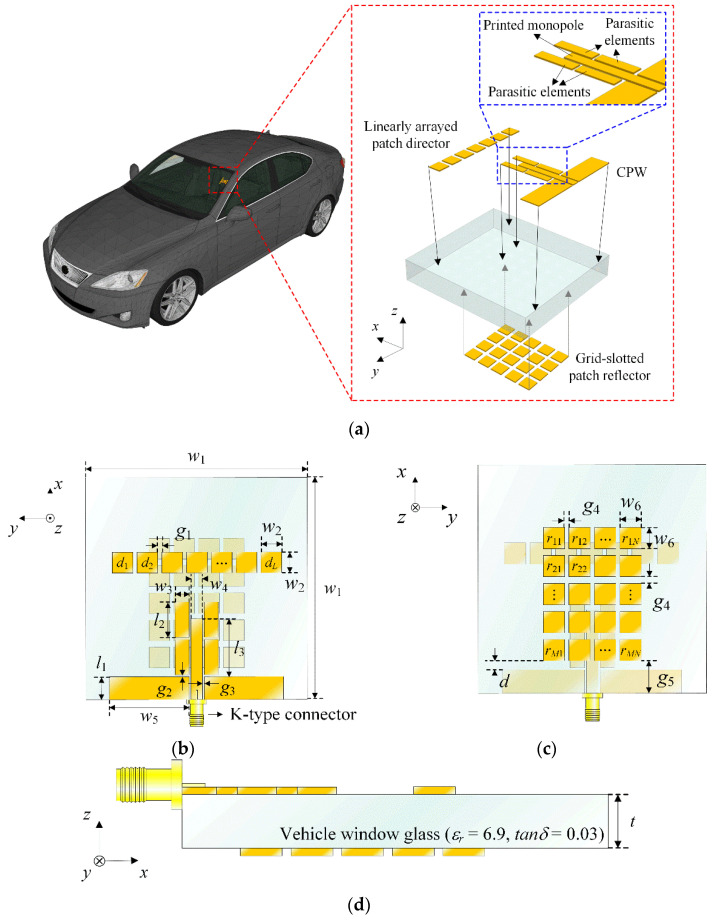
Geometry of the proposed antenna: (**a**) isometric view; (**b**) top view; (**c**) bottom view; (**d**) side view.

**Figure 2 sensors-22-06187-f002:**
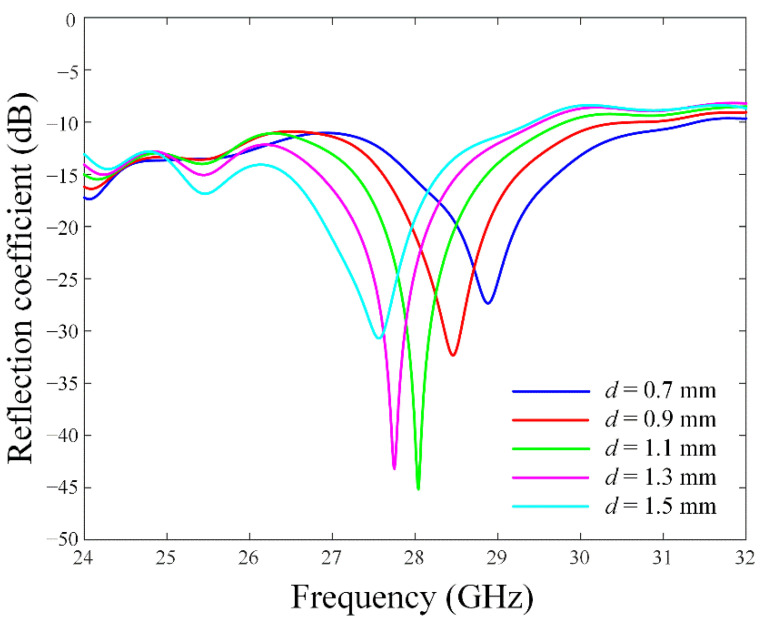
Reflection coefficients in accordance with the parameter *d*.

**Figure 3 sensors-22-06187-f003:**
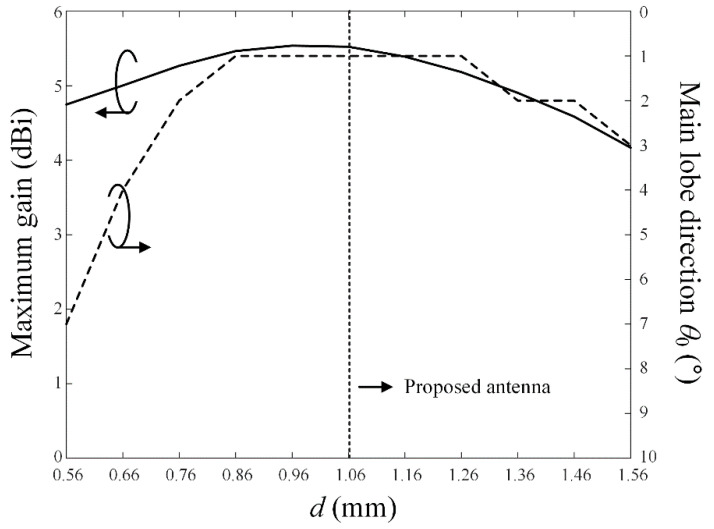
The maximum gain and main lobe direction *θ*_0_ in accordance with the parameter *d* at 28 GHz.

**Figure 4 sensors-22-06187-f004:**
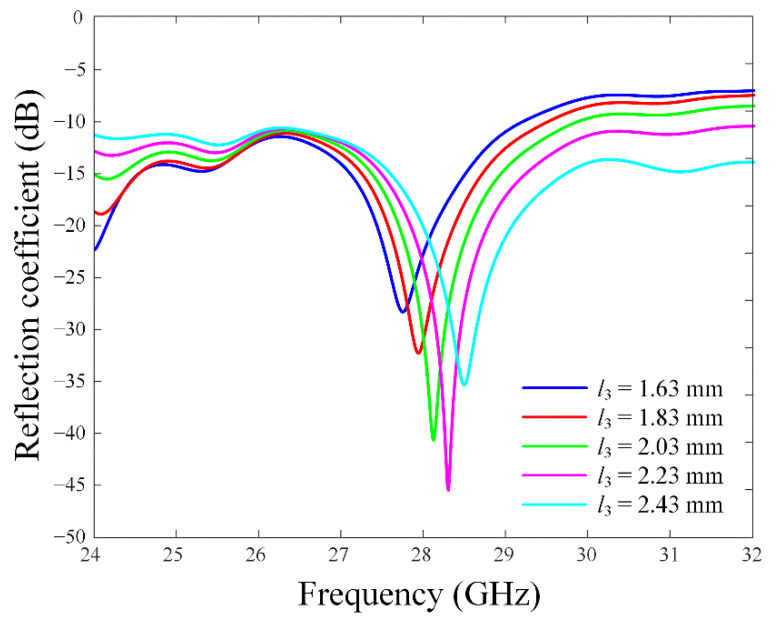
Reflection coefficients in accordance with the parameter *l*_3_.

**Figure 5 sensors-22-06187-f005:**
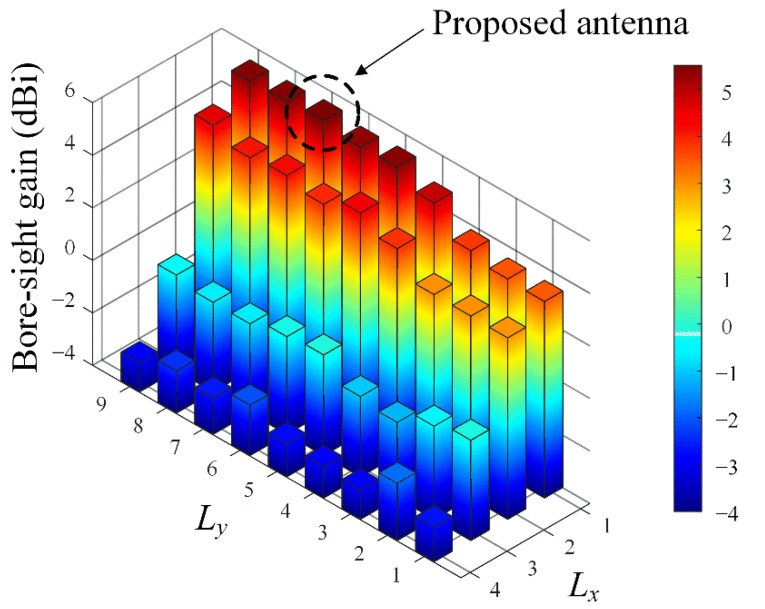
Bore-sight gain in accordance with the number of patch elements in the director.

**Figure 6 sensors-22-06187-f006:**
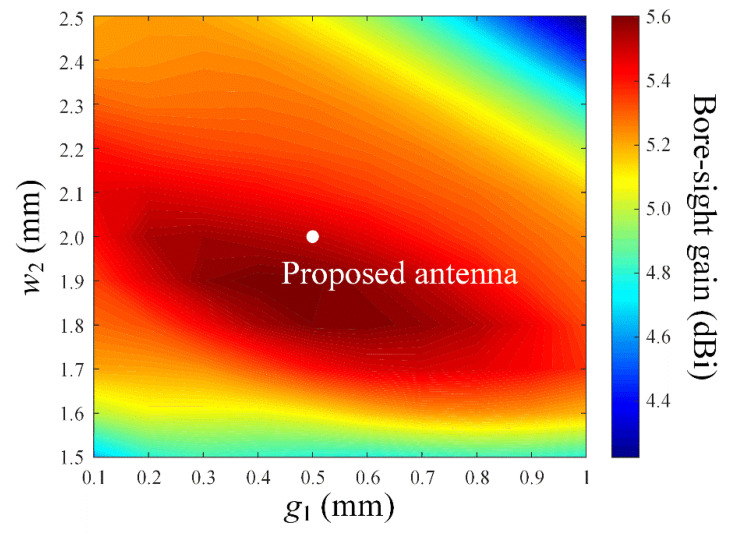
Bore-sight gain in accordance with parameters *g*_1_ and *w*_2_.

**Figure 7 sensors-22-06187-f007:**
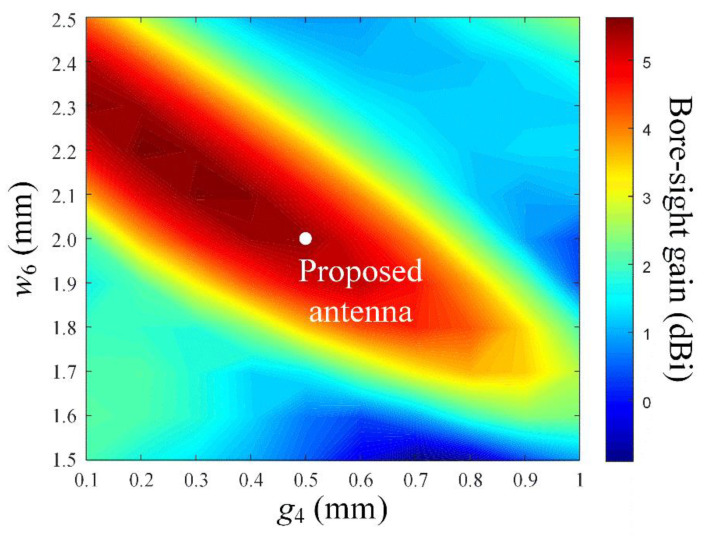
Bore-sight gain in accordance with parameters *g*_4_ and *w*_6_.

**Figure 8 sensors-22-06187-f008:**
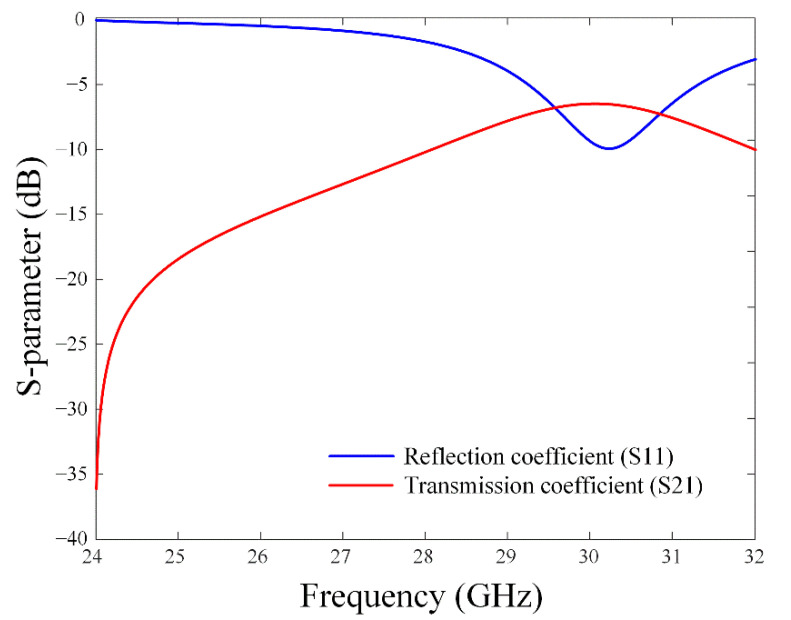
Reflection coefficient and transmission coefficient of the grid-slotted patch reflector.

**Figure 9 sensors-22-06187-f009:**
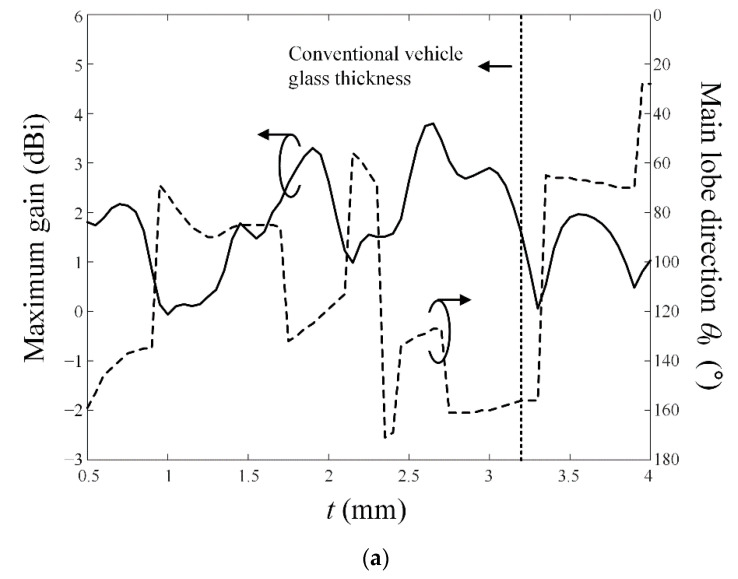
The maximum gain and main lobe direction *θ*_0_ in accordance with parameter *t*: (**a**) conventional printed monopole antenna; (**b**) proposed antenna.

**Figure 10 sensors-22-06187-f010:**
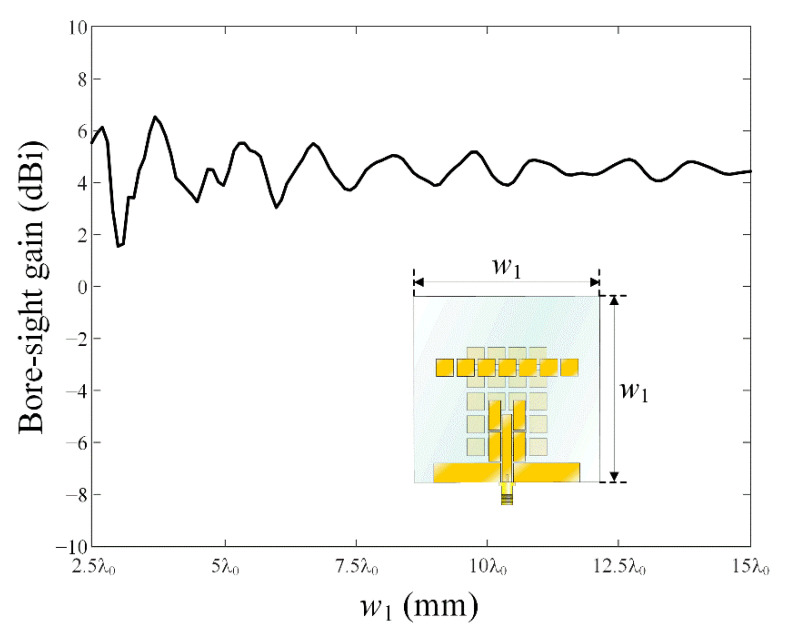
Bore-sight gain in accordance with the parameter *w*_1_.

**Figure 11 sensors-22-06187-f011:**
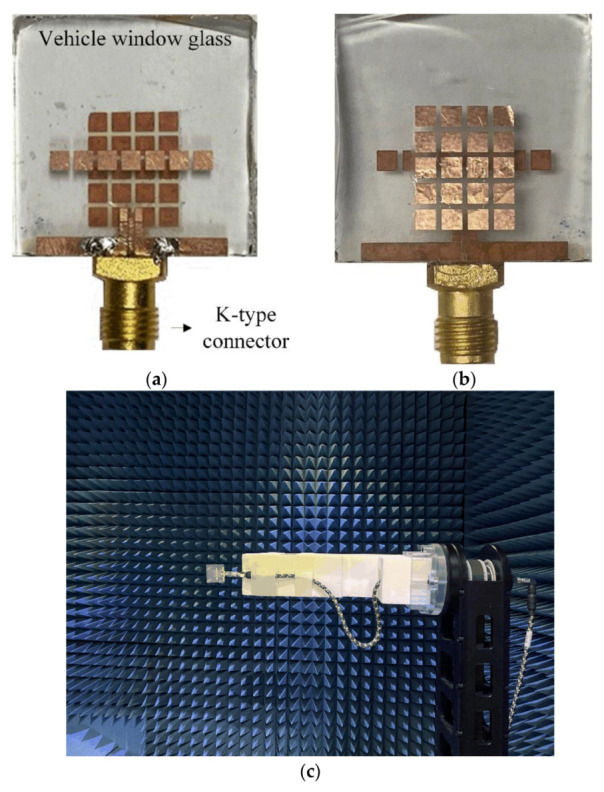
Photographs of the proposed antenna and measurement setup: (**a**) top view; (**b**) bottom view; (**c**) measurement setup.

**Figure 12 sensors-22-06187-f012:**
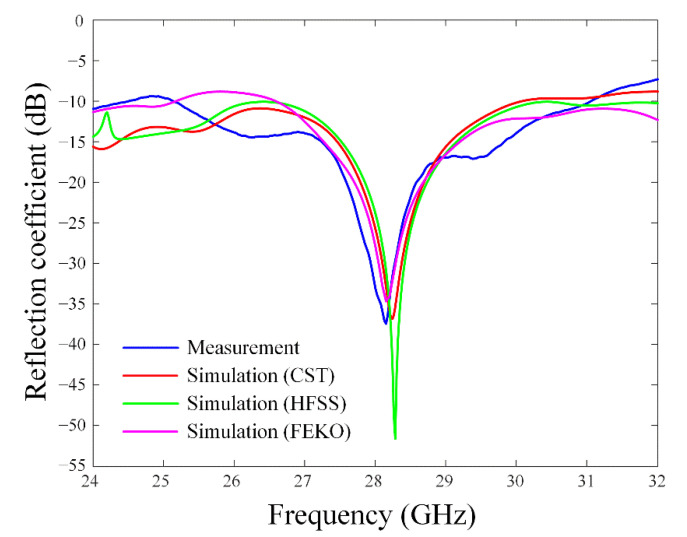
Measured and simulated reflection coefficients of the proposed antenna.

**Figure 13 sensors-22-06187-f013:**
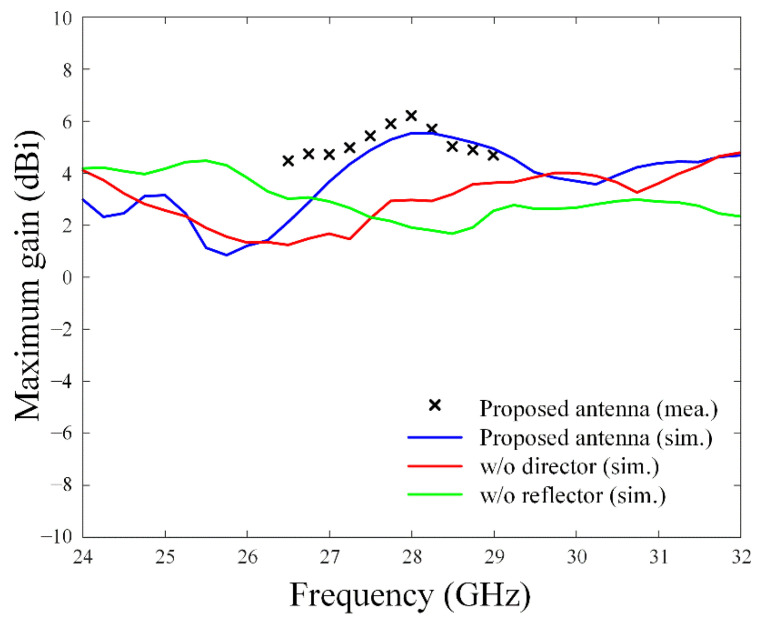
Maximum gains with and without the director and the reflector.

**Figure 14 sensors-22-06187-f014:**
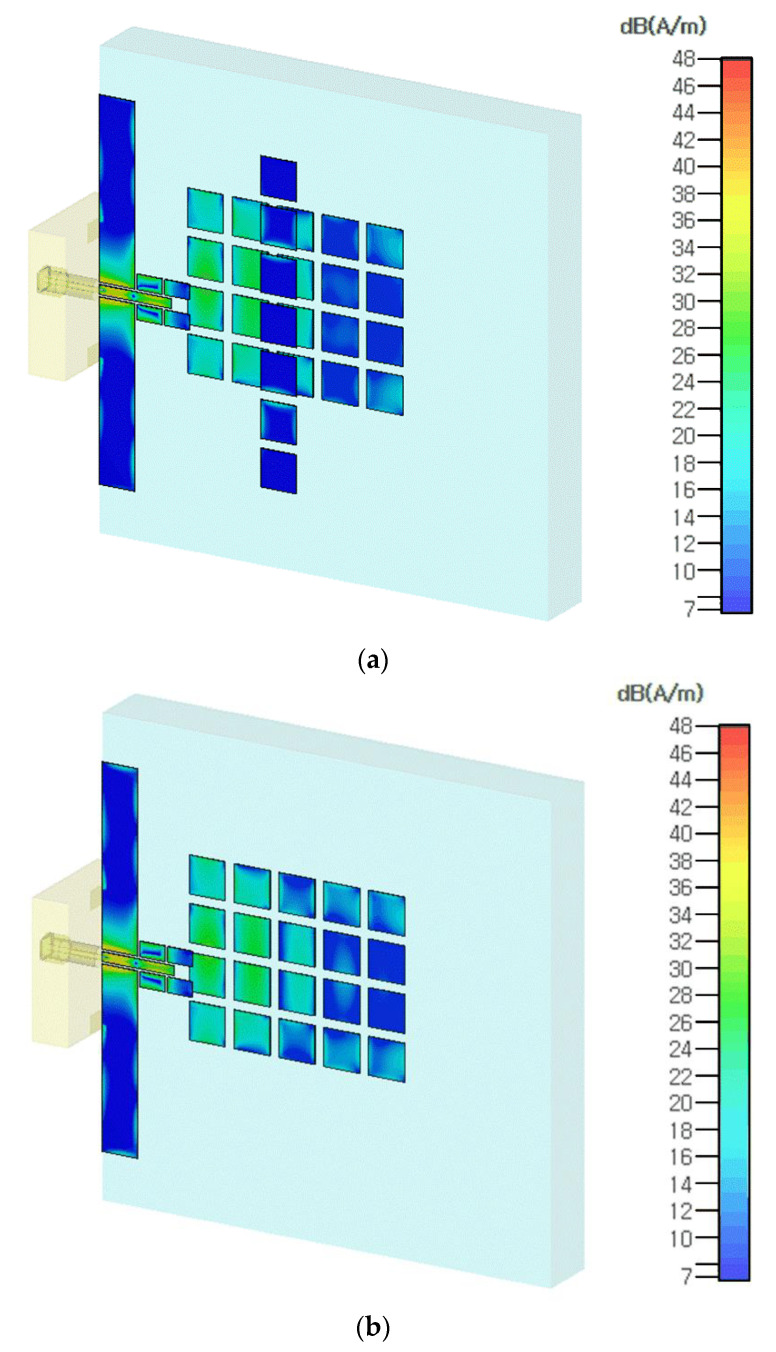
Current distributions: (**a**) proposed antenna; (**b**) without the director; (**c**) without the reflector.

**Figure 15 sensors-22-06187-f015:**
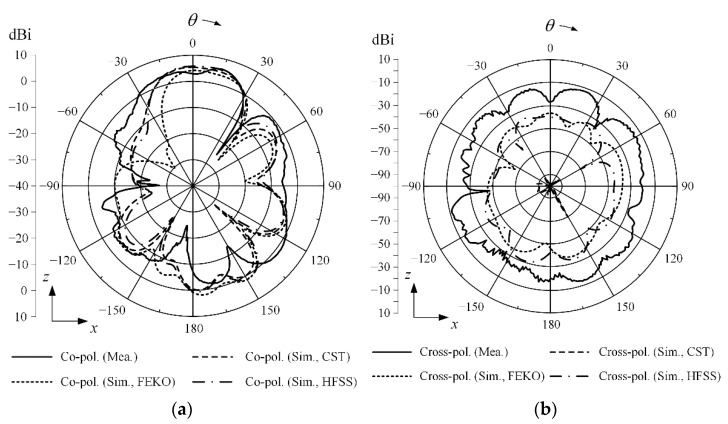
Measured and simulated 2-D radiation patterns of the proposed antenna at 28 GHz: (**a**) *zx*-plane co-polarization; (**b**) *zx*-plane cross-polarization; (**c**) *zy*-plane co-polarization; (**d**) *zy*-plane cross-polarization.

**Figure 16 sensors-22-06187-f016:**
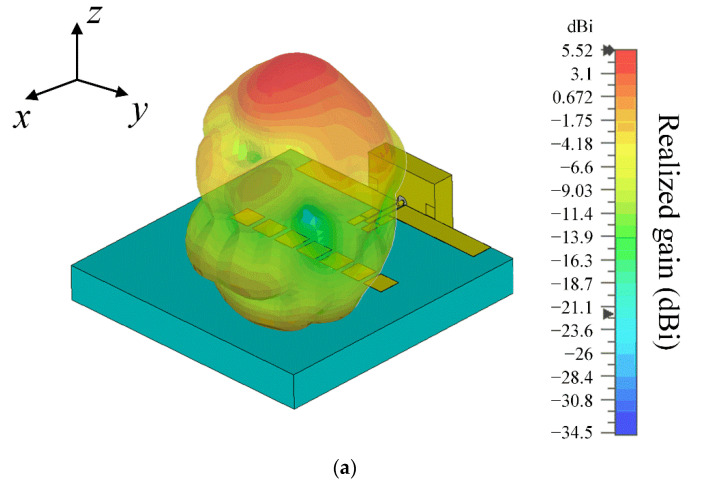
3-D radiation patterns by simulation: (**a**) proposed antenna; (**b**) without the director; (**c**) without the reflector.

**Table 1 sensors-22-06187-t001:** Design parameters of the proposed antenna.

Parameters	Values	Parameters	Values
*w* _1_	25 mm	*g* _1_	0.5 mm
*w* _2_	2 mm	*g* _2_	0.13 mm
*w* _3_	0.8 mm	*g* _3_	0.13 mm
*w* _4_	0.5 mm	*g* _4_	0.5 mm
*w* _5_	9.62 mm	*d*	1.06 mm
*w* _6_	2 mm	*L*	7
*l* _1_	2 mm	*M*	5
*l* _2_	1.4 mm	*N*	4
*l* _3_	2.03 mm	*t*	3.2 mm

## Data Availability

Not applicable.

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
