# Peer review of "Design of a mmWave Antenna Printed on a Thick Vehicle-Glass Substrate Using a Linearly Arrayed Patch Director and a Grid-Slotted Patch Reflector for High-Gain Characteristics"

_sensors, 2022, doi:10.3390/s22166187_

Round 1

Reviewer 1 Report

This paper proposed a mmWave antenna printed on a thick vehicle glass with a linearly arrayed patch director and a grid-shaped reflector for high-gain characteristics. The structure has a novelty for overcoming degradations due to the thick glass with high losses. The parametric studies based on full-wave simulations provide a good insight on the proposed antenna, and the prototype measurement data agrees well with the simulation. Here are some minor comments to improve the manuscript:

1. In Figure 1, the width of rectangular patch is w2, but in the description, it is incorrectly written as w_1. It should be corrected as “rectangular patches having a width of w_1” --> “rectangular patches having a width of w_2”

2. It is necessary to unify the terms. Please change the sentence as follows: "CPW feeding structure" --> "CPW transmission line"

3. It is needs to correct grammatical errors as follows: “a high-gain characteristics” --> “high-gain characteristics” “gain characteristics of the proposed antenna is observed” --> “gain characteristics of the proposed antenna are observed”

4. In Figure 7, it would be better to increase the resolution of the data. Maybe more observation points can show the tendency of the results more clearly.

5. It would be better to add a 3-D simulation radiation pattern of the proposed antenna to help understand the beam shape.

Author Response

We would like to thank the reviewers for their constructive comments and for taking the time to consider our paper. We have tried our best to revise the paper in accordance with the reviewers’ comments.

Reviewer 2 Report

The authors proposed the Design of a mmWave Antenna Printed on a Thick Vehicle Glass Substrate Using a Linearly Arrayed Patch Director and a Grid-shaped Reflector for High-Gain Characteristics. The idea is exciting, and the simulation results are reasonably good, showing potentially strong reconfigurability. However, the concept of this article is not understood. Here are some of my concerns listed below:

My Concern (about Antenna alone):

1- There is no monopole shape in the proposed antenna; from where did the author claim that there is a monopole shape??? The monopole antenna or monopole shape should be circular!

2- Where is the parametric study regarding the proposed antenna?

3- I can't see "The linearly arrayed patch director". It is a slot antenna with a rectangular shape!! 

4- Where is the operating bandwidth of the proposed antenna in the Abstract section?? The Abstract should be rewritten again?!

My Concern (Reflector sheet):

5- The proposed reflector is not a grid-shaped reflector!!! It is a square patch frequency selective surface (FSS) reflector (Capacitive reflector)!!

6- This FSS unit cell needs to be well studied in terms of transmission coefficient (S21) and reflection phase. We need to know why the gain increased after loading the FSS reflector.

7- The introduction is insufficient and needs to be improved. The authors are required to explain some of the Metamaterial reflectors and their performance in terms of gain and operational bandwidth. Frequency selective surface is considered one of them [1,2]; please see these articles, which may add value to the introduction:

1- A Wideband High-Gain Microstrip Array Antenna Integrated with Frequency-Selective Surface for Sub-6 GHz 5G Applications. Micromachines 202213, 1215. https://doi.org/10.3390/mi13081215.

2- Comparative Study of Square and Circular Loop Frequency Selective Surfaces for Millimeter-Wave Imaging Diagnostics Systems. Sensors 201818, 3079. https://doi.org/10.3390/s18093079.

8- Where are the parametric study of the proposed FSS reflector in term of the FSS unit cell and the gap between the antenna and the reflector?

9- Where is the simulated and measured gain with and without a reflector??

10- Where is the simulated and measured radiation efficiency with and without a reflector??

11- The figures are too boring. The authors should present a colour line to differentiate the curves and make it easy for the readers!

12- The authors should propose the current antenna distributions with and without reflectors??

That's all for me at this moment! The authors are required to revise the comments above carefully. Thanks

Author Response

(The authors gave the same response as above.)

Round 2

Reviewer 2 Report

The authors have revised the given comments successfully. However, typos and spacing errors still need to be carefully checked. Besides, please dont list the Figures one by one, such as those in Figure 2 up to 10.

Author Response

(The authors gave the same response as above.)
